# Selective Deoxygenation of Sludge Palm Oil into Diesel Range Fuel over Mn-Mo Supported on Activated Carbon Catalyst

Abdulkareem Ghassan Alsultan [1,*] , Nurul Asikin-Mijan [2,*], Laith K. Obeas [3], Aminul Islam [4] , Nasar Mansir [5] , Siow Hwa Teo [6] , Siti Zulaika Razali [7] , Maadh F. Nassar [8], Surahim Mohamad [8] and Yun Hin Taufiq-Yap [1,9,*]

[1] Chemistry Department, Faculty of Science and Natural Resources, Universiti Malaysia Sabah, Kota Kinabalu 88400, Sabah, Malaysia
[2] Department Of Chemical Sciences, Faculty of Science and Technology, Universiti Kebangsaan Malaysia Bangi, Bandar Baru Bangi 43600, Selangor Darul Ehsan, Malaysia
[3] Department Chemical Engineering, Technical Institute of Babylon, Al-Furat Al-Awsat Technical University (ATU), Al-Najaf City 51001, Iraq; laith1978@atu.edu.iq
[4] Department Of Petroleum and Mining Engineering (PME), Jashore University of Science and Technology, Jashore 7408, Bangladesh; aminul03211@yahoo.com
[5] Department Of Chemistry, Faculty of Science, Federal University Dutse, Dutse 7156, Jigawa, Nigeria; nmansir09@yahoo.com
[6] Industrial Chemistry Program, Faculty Science and Natural Resources, Universiti Malaysia Sabah, Kota Kinabalu 88400, Sabah, Malaysia; tony@ums.edu.my
[7] Nanomaterials Processing and Technology Laboratory, Institute of Nanoscience and Nanotechnology, Universiti Putra Malaysia Serdang, Seri Kembangan 43400, Selangor, Malaysia; zulaika@upm.edu.my
[8] Department Of Chemistry, Faculty of Science, Universiti Putra Malaysia Serdang, Seri Kembangan 43400, Selangor, Malaysia; nassarmaadh@gmail.com (M.F.N.); surahim88@gmail.com (S.M.)
[9] Vice Chancellor Office, Universiti Malaysia Sabah, Kota Kinabalu 88400, Sabah, Malaysia
* Correspondence: kreem.alsultan@yahoo.com (A.G.A.); ckin_mijan@yahoo.com (N.A.-M.); taufiq@upm.edu.my (Y.H.T.-Y.); Tel.: +60-182534058 (A.G.A.); +60-176545305 (N.A.-M.); +60-3-89466809 (Y.H.T.-Y.); Fax: +60-3-89466758 (Y.H.T.-Y.)

**Abstract:** Originating from deoxygenation (DO) technology, green diesel was innovated in order to act as a substitute for biodiesel, which contains unstable fatty acid alkyl ester owing to the existence of oxygenated species. Green diesel was manufactured following a process of catalytic DO of sludge palm oil (SPO). An engineered $Mn_{(0.5\%)}$-$Mo_{(0.5\%)}$/AC catalyst was employed in a hydrogen-free atmosphere. The influence of Manganese (Mn) species (0.1–1 wt.%) on DO reactivity and the dissemination of the product were examined. The $Mn_{(0.5\%)}$-$Mo_{(0.5\%)}$/AC formulation gave rise to a superior harvest of approximately 89% liquid hydrocarbons; a higher proportion of diesel fraction selectivity $n$-($C_{15}$+$C_{17}$) was obtained in the region of 93%. Where acid and basic active sites were present on the $Mn_{(0.5\%)}$-$Mo_{(0.5\%)}$/AC catalyst, decarboxylation and decarbonylation reaction mechanisms of SPO to DO were enhanced. Evidence of the high degree of stability of the $Mn_{(0.5\%)}$-$Mo_{(0.5\%)}$/AC catalyst during five continuous runs was presented, which, in mild reaction conditions, gave rise to a consistent hydrocarbon harvest of >72% and >94% selectivity for $n$-($C_{15}$+$C_{17}$).

**Keywords:** activated carbon; deoxygenation; heterogeneous catalyst; green diesel; biofuels

## 1. Introduction

Owing to current consternation regarding environmental and sustainability issues, renewable fuels have gained considerable attention. There is a significant requirement for fuel in the transportation domain; however, the manufacture of biofuel necessitates numerous feed sources. At present, large quantities of comestible feeds are utilised in order to diminish greenhouse gas emissions relative to those produced by liquid fossil fuels [1,2]. Nevertheless, such feeds are also a requisite to fulfil the nutritional needs of an expanding populace; when such food crops are diverted to fuel synthesis, their costs increase [3,4].

The interruption in food sources can be circumvented by the use of inedible oleaginous vegetation, which lives in dry soil [5]; however, their production would require upscaling to meet demand. In comparison to terrestrial plants, a greater yield of oil can be obtained from microalgae, but energy requirements and expense limit their use [6]. An alternative in comestible feeds is therefore necessary in order to fulfil the requirements for fuel; fats, oils and grease (FOG) waste streams appear extremely encouraging in this regard as they are inexpensive and widely accessible.

Significant oleaginous waste streams encompass yellow grease (YG), i.e., discarded cooking oil, and brown grease, which contains FOG gathered from grease traps, i.e., apparatus designed to segregate insoluble oils from the wastewater streams from industrial kitchens [7]. Despite the fact that a notable proportion of YG is presently transformed into renewable fuels, trap grease, i.e., sludge oil palm oil (SPO) obtained from the palm oil industry, is insufficiently used for this purpose. In fact, it was gauged by the United States (US) Department of Energy that annually, the US generates 1.7 million tons of trap grease, which undergoes incineration or is incarcerated in landfills. Thus, promoting the value of this product is an attractive proposition.

Specifically, the elevated oxygen concentration within biodiesel and bio-oils underlies the majority of their disadvantages, such as lower heat properties, when judged against their traditional fossil-derived counterparts. The manufacture of oxygen-free hydrocarbon from the catalytic deoxygenation (DO) of fatty acids and associated compounds has caught the eye of numerous workers in the field. In theory, DO entails the eradication of oxygen from the above substances via decarboxylation ($-CO_2$) and decarbonylation ($-CO$, $H_2O$) (deCOx) reactions within a hydrogen ($H_2$)-free environment. Furthermore, both green and petroleum-derived diesels have physicochemical characteristics in common, and so the former can be utilised alone or admixed with traditional diesel in combustion ignition engines without any engine adaptations.

Currently, there are numerous diverse solid catalysts that have been suggested for the manufacture of green fuel owing to their straightforward decontamination phase and recycling ability [8]. These are divided into two principal classes: (i) basic, i.e., alkaline-based and alkaline-earth-based metal oxides; and (ii) acid, e.g., functionalised silica substances, MCM-41 and SBA-15, with organic-sulphonic acids [9]. In general, in contrast to the solid acid catalysts, the DO reaction using solid base catalysts showed mild reactivity in the context of a strong reaction [10]. Nevertheless, SPO is an acid oil containing high quantities of free fatty acids (FFAs); the active moieties of the base catalyst utilised for the DO reaction are generally inhibited by the FFA as a consequence of a saponification event [11].

An acid catalyst is thought to be a more favoured contender for the concurrent DO of FFAs and triglycerides in order to attain a single-pot admixture of alkane and alkene from SPO at elevated temperatures [12]. A heterogeneous solid acid catalyst is, therefore, an environmentally friendly, cost-efficient option for green fuel manufacture from poor quality oil at cooler temperatures [13–17]. Nevertheless, the principal issue to overcome is to design an efficacious solid acid catalyst for the DO reaction when mild reaction contexts are enduring. A significant amount of earlier research relating to deCOx has concentrated on the Pd- and Pt-supported catalysts, which demonstrated good transformation and differentiation of diesel-type hydrocarbons [7,10–21], but the price of the precious metals has restricted the commercial upscaling of their utility. In particular, lower-cost carbon-based catalysts have shown practically analogous outcomes to those relating to the Pd- and Pt-associated compounds for transforming lipid-founded feeds into fuel-related hydrocarbons [18,19]. Indeed, Manganese (Mn) and molybdenum (Mo) were found to be effective promoters of DO reactions. However, few studies have described supported Mo and Mn during the DO process [20,21]. In this work, a series of Mn–Mo bimetallic catalysts loaded on acidic supports is focused on the DO of SPO. As far as the researcher is aware, this is the first paper which documents the investigation of a non-sulphide Mn/Mo catalyst in the previously described circumstances which is permitted to achieve a steady state, as well as being the first appraisal of the stability and utility of such a catalyst.

## 2. Results and Discussion

### 2.1. Deoxygenation Reaction Parameters Optimization

The appearances of the solid catalysts were additionally appraised in order to explore their configuration, pattern and surface particle contours. FESEM images for the made-up specimens were evaluated and demonstrated a homogeneous allocation of metal on the support (Figure 1a–c). The dimensions and form of the particles were notably altered following active metal doping. It can be seen that the preparation context has a marked impact on the carbon's appearance. The dimensions of the particles were diminished from 27 nm to 312 nm on the heated impregnated bimetallic catalysts by elevating the Mn and Mo doping proportion from 0.1 to 1 wt.%. In contrast, raising Mn to >0.5 wt.% was associated with particle enlargement from 92 nm to 312 nm owing to aggregation effects. These data were validated utilising BET and XRD, which demonstrated a reduction in surface area and volume of the pores, together with an augmentation in crystal dimensions. Catalysts which exhibit superior activity generally have a reasonable surface area and particle size; the latter, configuration and porous nature, have a significant impact on the reaction. Thus, the minute particle size of organised porous support would enhance surface area and generate a greater active catalytic surface, properties that would be advantageous for reaction performance [22,23]. The surface area was increased by increasing the element loading from 0.1 to 1 %, which is due to the metal sate acting as an activation agent to the activated carbon. On the other hand, the pore volume and size were reduced by increasing the doping percentage due to the fact that the active element will block the pores. Catalysis science has demonstrated the importance of the active site to the catalyst's activity [24].

The X-ray diffraction (XRD) pattern of AC-Sr evidenced no crystalline $Sr_2O$ phases (2a), suggesting that Strontium is either highly dispersed on the AC external surface or doped into the pore (Figure 2). Diffraction bands indicating the cubic metallic Mn and Mo reflections with 2θ: 43.47°, 46.03°, 48.37°, 50.67°, 52.55°, 54.93°, 62.72°, 66.22°, 77.54° and 79.07° with *hkl* values of: (300), (310), (311), (222), (320), (321), (330), (420), (510) and (511) (Reference code 00-003-1014) with fcc at 48.37°, and 2θ: 38.66°, 44.95°, 65.45°, 78.68° and 82.92° with *hkl* value of: (111), (200), (220), (311) and (222) (Reference code:01-088-2331) with space groups of Fm-3m, respectively. Corresponding Mn-Mo/Ac catalyst crystalline sizes were calculated according to the Scherrer equation as 13.9, 15.7 and 8.5 $Mn_{(0.5\%)}$-$Mo_{(0.5\%)}$/AC, $Mn_{(0.5\%)}$-$Mo_{(0.1\%)}$/AC and $Mn_{(0.5\%)}$-$Mo_{(1\%)}$/AC, respectively, suggesting the introduction of Sr inhibited the sintering of Mn and Mo nanoparticles.

The contrast of the TPD-$NH_3$ data sets for the Mn/AC and Mn-Mo/AC catalyst specimens implied that the Mn support predominantly generated robust acid loci within Mn-Mo/AC (Figure 3). The AC support on the Mn-Mo specimen evidenced an asymmetric data set of low acidity below a temperature of 550 °C, which was in keeping with the characteristics of the earlier-generated AC supported catalysts utilised for alternative purposes [25]. In general, the TPD-$NH_3$ parameters of the catalyst specimens suggested that there was a robust engagement between the AC support and the two metals. The most significant variations observed between the Mn/AC and Mn-Mo/AC specimens were associated with alterations in the acidity scale. This predominantly arose from the presence of Mo in Mn/AC. It can be gauged that in the process of loading with Mo, each acid has $6H^+$ medium loci that are swapped with a single $Mo^{+6}$ ion to maintain charge equilibrium, i.e., $6H^+/Mn^{+6} = 1$ [26]. This will give rise to the generation of a de novo $NH_3$ desorption zenith, i.e., the robust acid loci on the Mn-Mo/AC catalyst. Furthermore, the minimal alterations seen in the temperature sites and the strengths of the medium and concentrated acid loci in the spectra after Mn loading at 0.5 wt.% implied that the quantity and strength of the acid loci were conserved and thus maintained the integrity of the catalyst's performance for the typical DO reaction of SPO. Mn-doping enhanced the acid loci density, suggesting that Mo has higher acidity than Mn/AC.

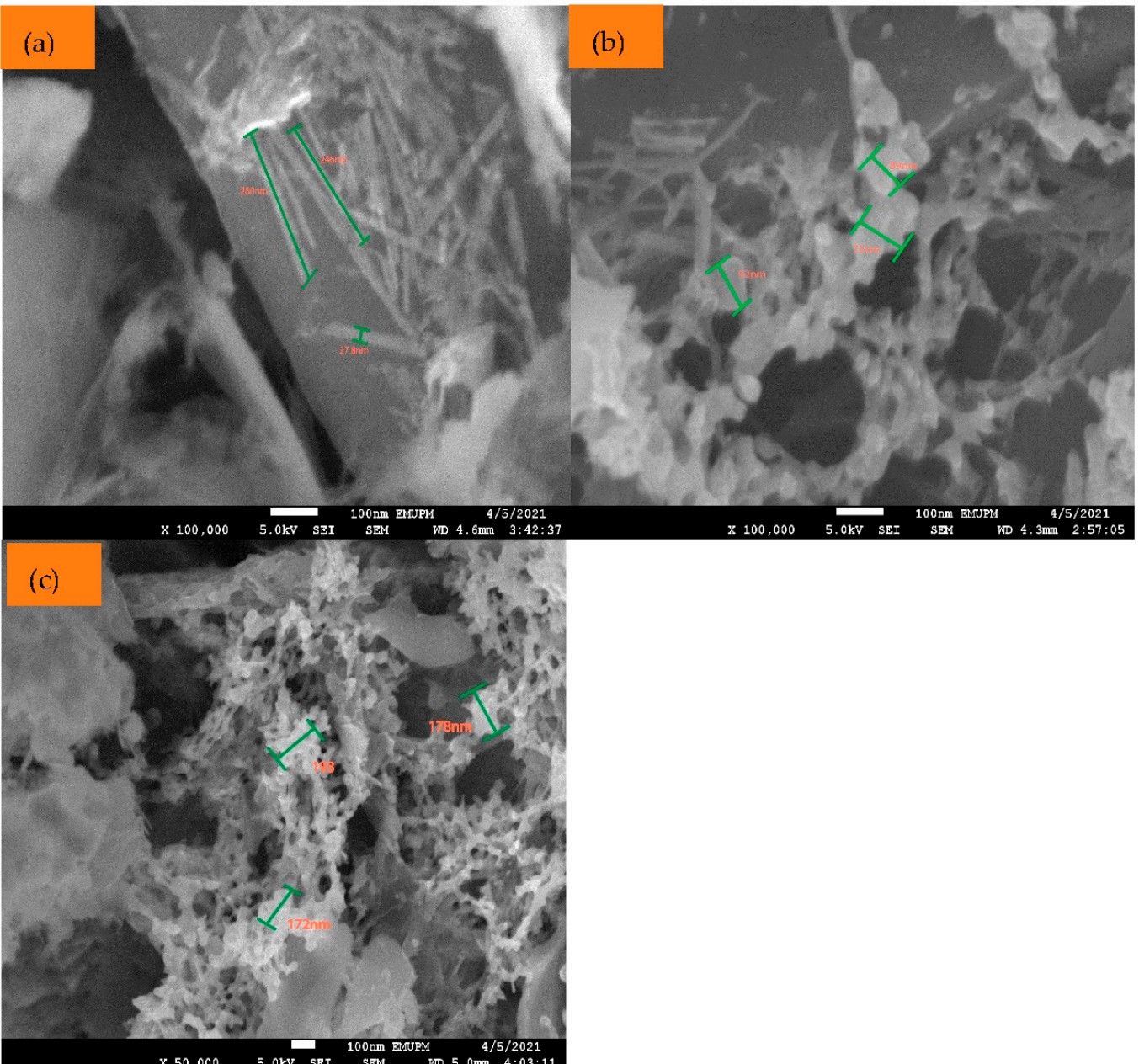

**Figure 1.** Representative FESEM images of samples. (**a**) Mn$_{(0.5\%)}$-Mo$_{(0.1\%)}$/AC, (**b**) Mn$_{(0.5\%)}$-Mo$_{(0.5\%)}$/AC and (**c**) Mn$_{(0.5\%)}$-Mo$_{(1\%)}$/AC.

To determine the electronic states of Mo and Mn, X-ray photoelectron spectroscopy (XPS) measurements were carried out on the obtained metal states (Figure 4). The high-resolution spectrum cantered on Mo$^{+4}$3d$_{5/2}$ and Mo$^{+6}$3d$_{5/2}$/3d$_{3/2}$ can be deconvoluted into three contributions positioned at 228.9 eV, 232.1eV and 235.3 eV corresponding to Mo$^{+4}$3d$_{5/2}$ and Mo$^{+6}$3d$_{5/2}$/3d$_{3/2,}$ respectively(Figure 4A). The high-resolution spectrum cantered on Mn can be deconvoluted into two contributions positions at 653.70 eV and 642.05 eV (Figure 4B). The Mn 2p spectra were placed at the same minimum background level and normalised to the maximum intensity of the higher peak in order to make the comparison easier. The Mn 2p spectra of the Mn$_{(0.5\%)}$-Mo$_{(0.1\%)}$/AC catalyst were practically identical, with the Mn 2p$_{1/2}$ peak maximum located in the 642.2–642.25 eV region and the Mn 2p3/2 peak maximum located in the 653.70−654.5 eV range. This synergic effect between Mn and Mo will lead to the enhancement of the catalyst's acidic properties, which will enhance the catalytic deoxygenation activity [27,28].

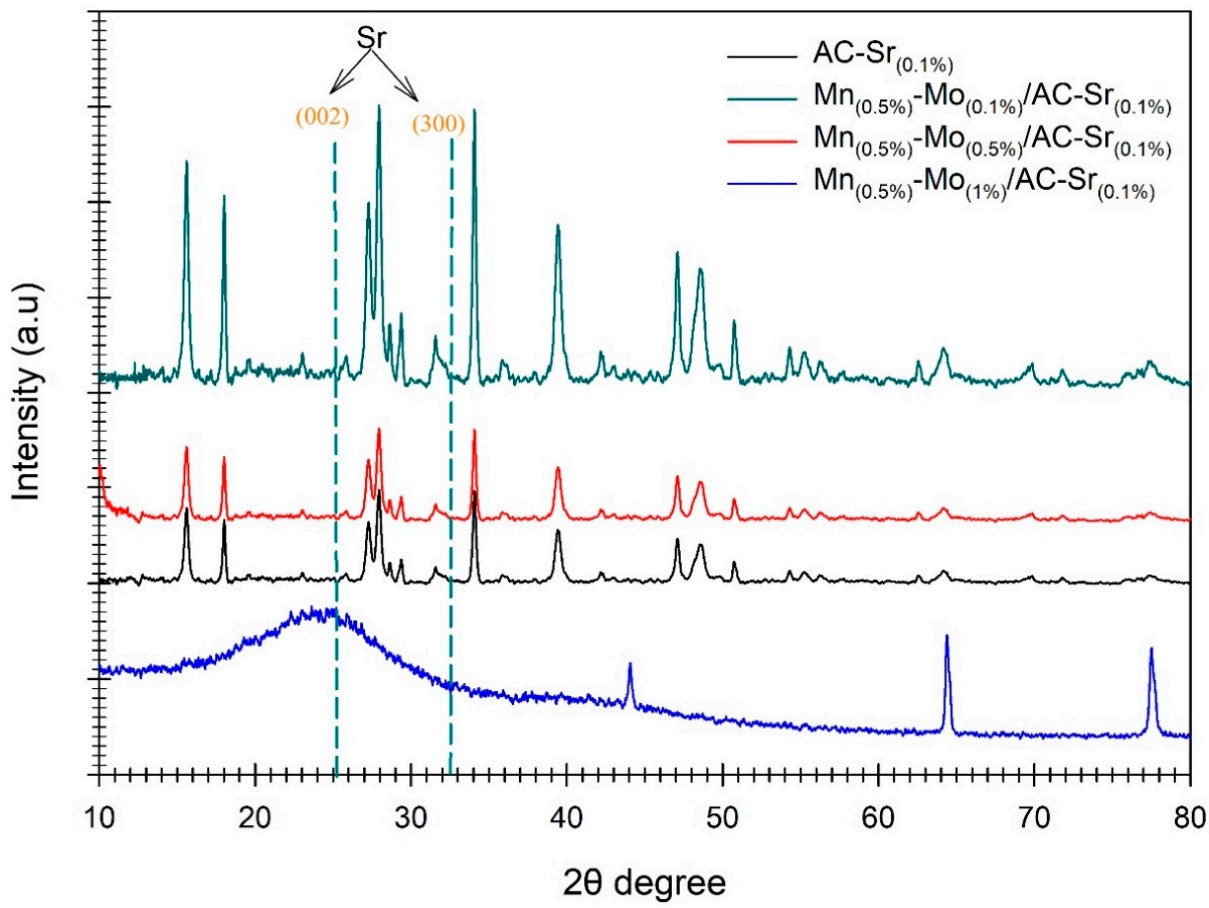

**Figure 2.** Structural properties of supports and catalysts. a XRD patterns of AC and Mn$_{(0.5\%)}$-Mo$_{(0.5\%)}$/AC, Mn$_{(0.5\%)}$-Mo$_{(0.1\%)}$/AC and Mn$_{(0.5\%)}$-Mo$_{(1\%)}$/AC.

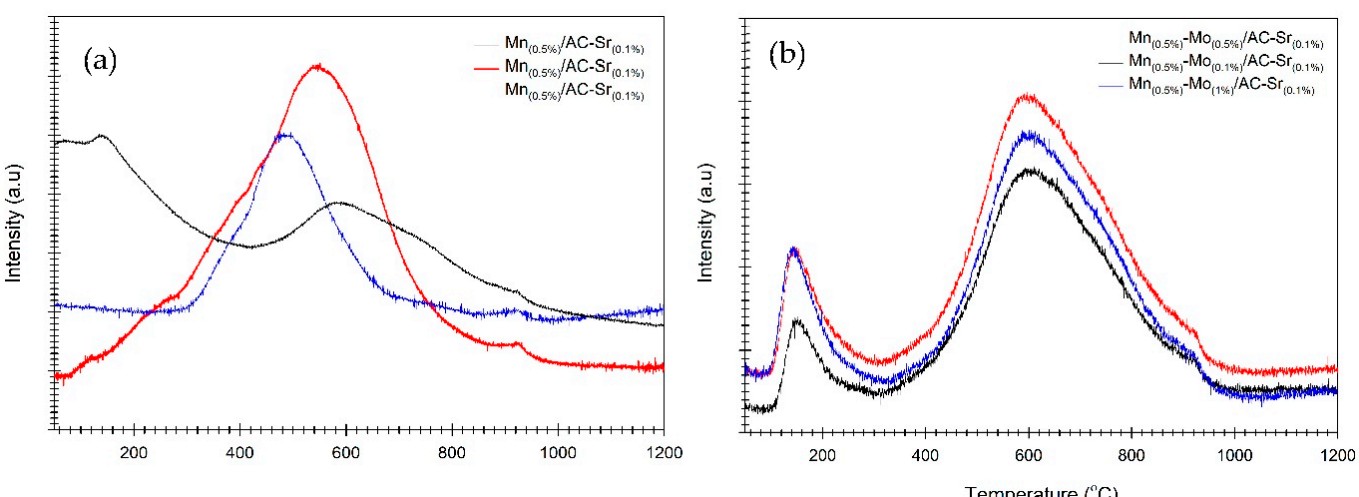

**Figure 3.** Acidity measurements of supported catalyst of (**a**) Mn/AC and (**b**) Mn-Mo/AC.

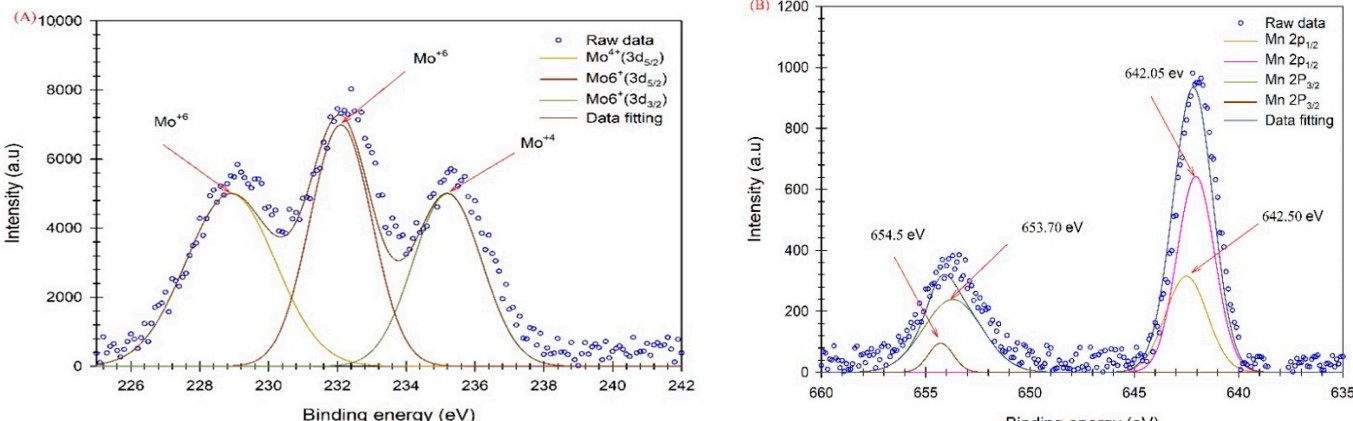

**Figure 4.** X-ray photoelectron spectroscopy (XPS) analysis of $Mn_{(0.5\%)}$-$Mo_{(0.1\%)}$/AC. (**A**) Molybdenum (Mo) oxidation statuses and (**B**) Manganese (Mn) oxidation statuses.

### 2.2. Catalytic Deoxygenation Reaction of SPO

When SPO underwent the DO reaction on Mn/AC and Mn-Mo/AC catalyst specimens, the products encompassed liquid *n*-alkanes and alkene, i.e., *n*-$C_7$–*n*-$C_{20}$, and oxygenated intermediate substances, e.g., alcohol, aldehyde and ketones, together with vapourised CO and $CO_2$. The SPO–$CO_2$ pathway generates *n*-$C_{17}$; oxygen was subtracted in the form of $H_2O$, and/or alcohol was generated via the –$CO_2$/–CO pathway of SPO and the oxygenated intermediates, with the oxygen being eliminated as CO. The manufactured catalysts produced with either alone or two metal catalysts, i.e., Mn/AC and Mn-Mo/AC, offered the superior transformation and total differentiation of *n*-$C_{17}$ hydrocarbon during the DO of SPO. The performance of the catalyst for the reaction was predominantly a consequence of Mn and Mo surface density and the acidity of the catalyst. The DO reaction was conducted for 30 min at a reaction temperature of 300 °C with an agitation speed of 500 rpm within an inert $N_2$ atmosphere. The screening of the Mn and Mo loading percentage on the activated carbon show that loading the activated carbon by 0.5 wt.% increases the hydrocarbon yield from 31.9% to 37.9% compared to 0.1 wt.% and increases the Mo loading from 0.1 wt.% to 0.5 wt.% leading to an increase in the yield from 25.7% to 42.4%. The screening results show the best Mn loading percentage was 0.5 wt.%, and this percentage was selected. The effect of Mo on the Mn catalyst was studied from 0.1 to 1 wt.% loading percentage, and the results show the 0.5 wt.% of Mn and Mo was the best percentage, with a hydrocarbon yield of 74% (Figure 5).

### 2.3. Optimisation Study of the Reaction Parameters

Pilot work demonstrated that the bimetallic catalyst specimen, $Mn_{(0.5\%)}$-$Mo_{(0.5\%)}$/AC, was the most efficacious for DO for SPO. Thus, it was utilised in order to evaluate the outcome for varying reaction indices, i.e., catalyst loading, temperature and time, on the hydrocarbon transformation and selectivity.

#### 2.3.1. Effect of Catalyst Loading

The impact of catalyst loading on hydrocarbon transformation and the differentiation of products is presented in Figure 6a,b. The DO reaction was conducted for 30 min at a reaction temperature of 300 °C with an agitation speed of 500 rpm within an inert $N_2$ atmosphere. The impact of catalyst loading, i.e., 1–6 wt.%, was evaluated. The data demonstrated that DO performance and *n*-$C_{15}$+$C_{17}$ differentiation were both elevated over a catalyst loading range of 1–2 wt.%, but then fell following catalyst loads >2 wt.%. HC yield and product selectivity for *n*-$C_{17}$ were superior, i.e.,47.13 % and 72.85%, respectively, when 2 wt.% was employed. In contrast, a drop in DO performance and *n*-$C_{15}$+$C_{17}$ selectivity at catalyst loads over 2 wt.% may reflect overburdening of the catalyst, which will promote a drive towards secondary and concurrent reactions owing to the surplus active loci for

different reactions [29]. Employment of 2 wt.% of catalyst alone was noted to be efficacious and cost-effective.

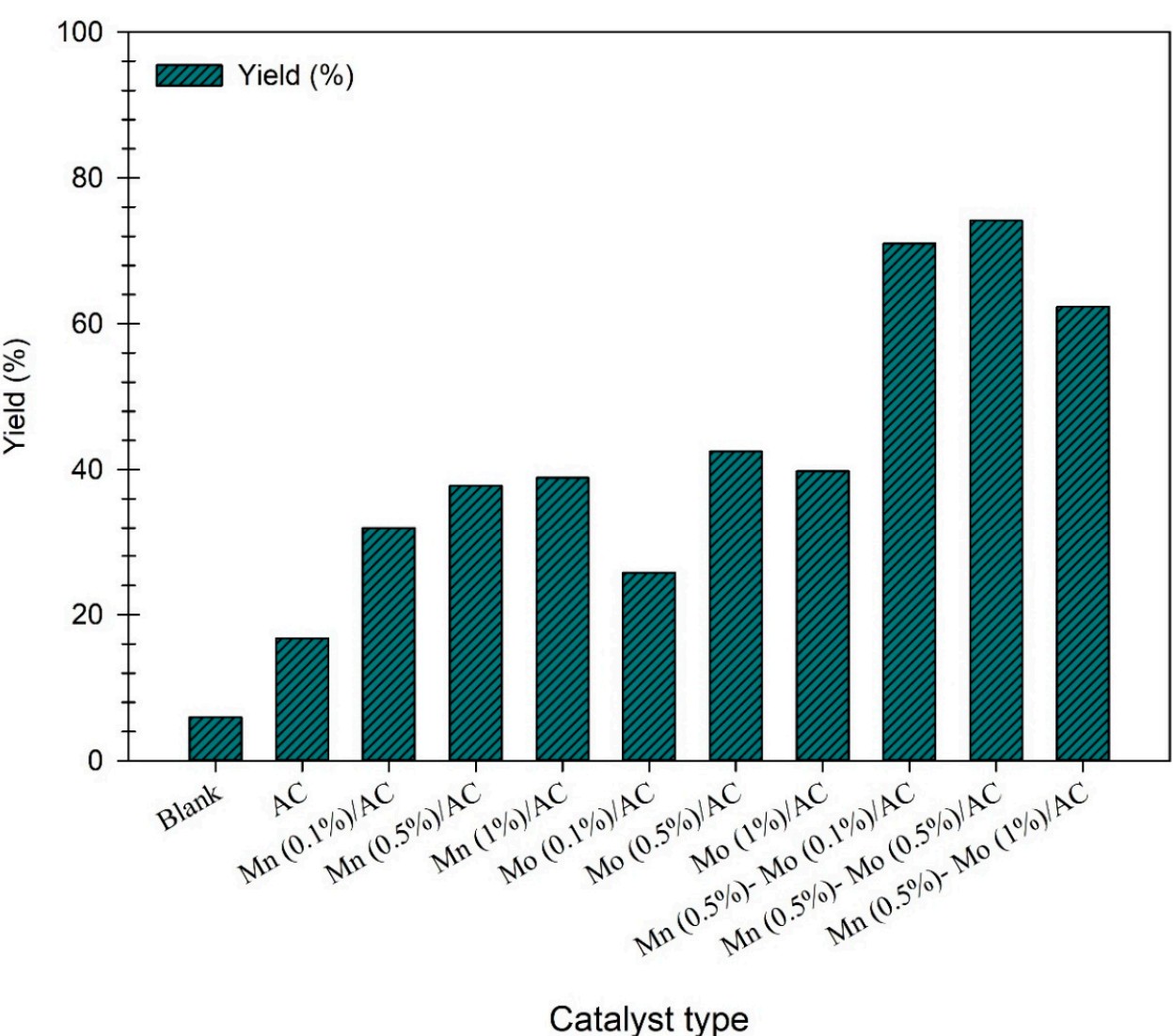

**Figure 5.** Deoxygenation reaction screening of SPO over supported catalyst.

2.3.2. Effect of Reaction Time

The impact of reaction time on SPO DO performance was assessed (Figure 6c,d). The DO was conducted within a spectrum of reaction times, i.e., 30–240 min at a temperature of 300 °C. Catalyst loading was 0.5 wt.%, agitation speed was 500 rpm and the reaction was performed in an inert atmosphere. Hydrocarbon transformation rose with the prolongation of the reaction time as anticipated. Both the above and the required $n$-$C_{15}$+$C_{17}$ selectivity became notably elevated from the first 30 min to 60 min, with rises from 29.72% to 64.27% and 55.93% to 87.84%, respectively. This implied that prolonged reaction times could promote DO performance through deCOx pathways by enhancing the capacity of the involved compounds to react with the surface of the catalyst. Above the 60 min reaction time, $n$-$C_{15}$+$C_{17}$ hydrocarbon transformation diminished to 73%, which may be a result of the oxygenated liquid product cracking into lower weight segments, which, in turn, may give rise to the generation of vapourised substances and the deactivation of the active loci on the catalyst through a side reaction, i.e., an active locus with CO and $CO_2$ [30].

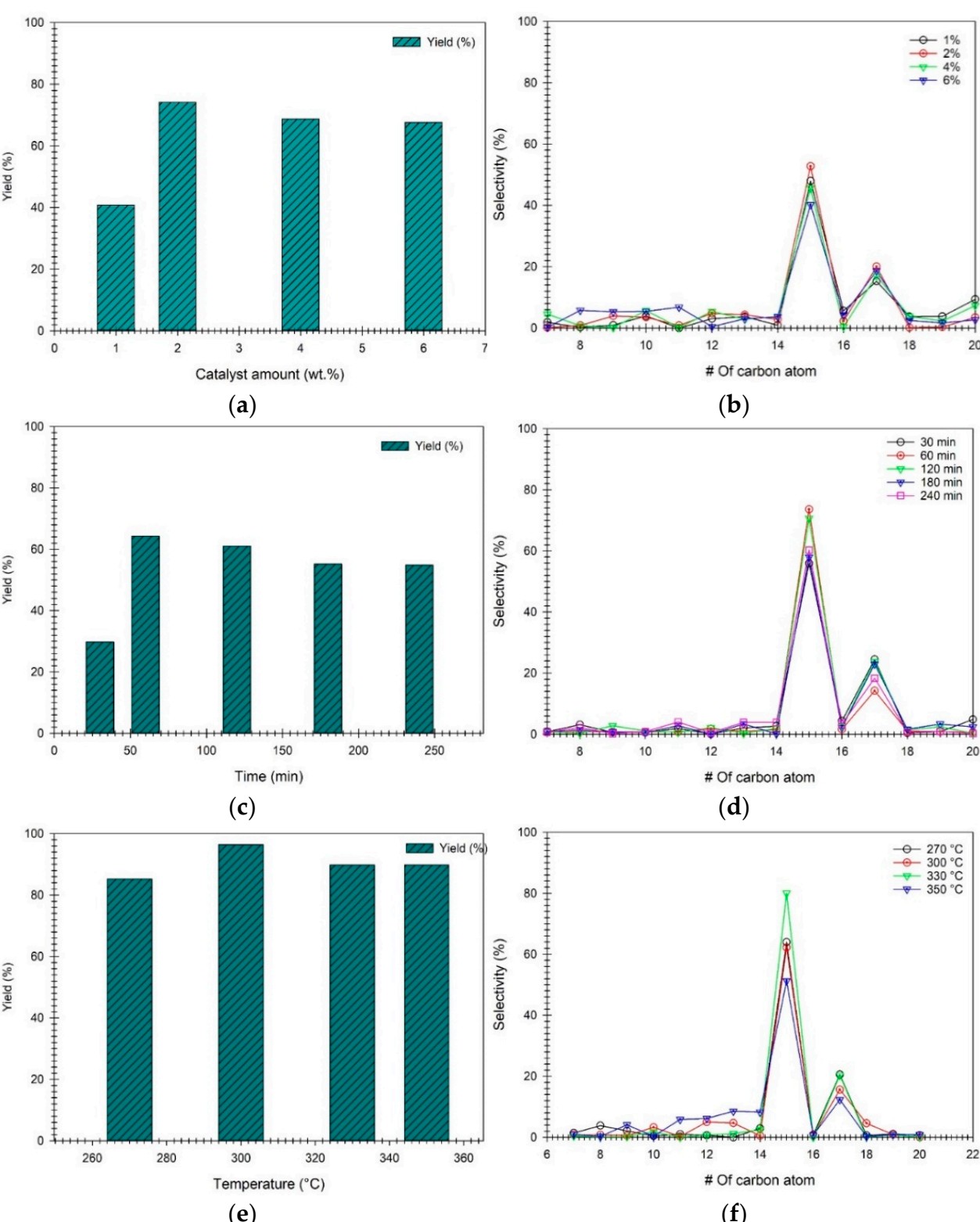

**Figure 6.** Reaction optimization condition of $Mn_{(0.5\%)}$-$Mo_{(0.5\%)}$/AC. (**a,b**) Effect of catalyst loading, (**c,d**) effect of reaction time and (**e,f**) effect of reaction temperature.

### 2.3.3. Effect of Temperature

The consequence of temperature, i.e., within the spectrum of 270 °C to 350 °C, on the SPO DO reaction was studied (Figure 6e,f). The reaction was conducted utilising catalyst loading of 2 wt.%, agitation speed of 500 rpm and 60 min reaction time in an

inert environment. It became transparent that elevating the reaction temperature from 270 °C to 330 °C augmented hydrocarbon transformation from 85.22% to 96.37%. This was modestly diminished to 87.00% when the temperature of the reaction was raised to 350 °C. A comparable pattern was also evident rregarding the *n*-$C_{15}$+$C_{17}$ (94.44 % selectivity) differentiation, with the highest degree seen at a temperature of 330 °C and a fall to 89.77% when the temperature was raised to 350 °C. It was reported that a higher content of FFA in the feedstock will lead to a decrease in DO activity and favour the cracking reaction [31]. In this work, we studied the catalyst activity using fresh palm oil (FFA < 0.05%) at the optimum reaction conditions, the DO results show that the hydrocarbon yield using the fresh pam oil was increased from 96.37% for the sludge palm oil to 98.19% for the fresh palm oil (with increasing in the hydrocarbon yield of 1.82%) (Figure S1). This suggested that the $Mn_{(0.5\%)}$-$Mo_{(0.5\%)}$/AC were stable at high FFA content feedstocks.

### 2.4. Catalyst Reusability Study

A DO reaction in the batch reactor system state was examined in order to assess the reusability profile of the $Mn_{(0.5\%)}$-$Mo_{(0.5\%)}$/AC catalyst; conditions included 2 wt.% catalysts and a temperature of 330 °C over an hour. When an individual cycle was concluded, reactivation of the catalyst was carried out by rinsing in hexane; it could then be reutilised for the following cycle. Consistent activity of the $Mn_{(0.5\%)}$-$Mo_{(0.5\%)}$/AC catalyst was observed for five serial runs. A hydrocarbon yield of >94%, including *n*-$C_{15}$+$C_{17}$ products with a selectivity of >70%, was achieved (Figure 7a). These data suggest that the requisite mechanical attributes and chemical stability are present in the $Mn_{(0.5\%)}$-$Mo_{(0.5\%)}$/AC catalyst. There was a modest degree of leaching of $Mn^{3+}$ and $Mo^{3+}$ species from the $Mn_{(0.5\%)}$-$Mo_{(0.5\%)}$/AC catalyst within each cycle; this was confirmed by the elemental assessment. The change in quantities of $Mn^{3+}$ and $Mo^{3+}$ leaching into the liquid product from the original to the final run was determined by comparative analysis to slowly rise from 1 ppm to 10 ppm for $Mn^{3+}$ and from 2 ppm to 8 ppm for $Mo^{+3}$. The attrition in DO activity can therefore be attributed to the loss of the active metals into solution during serial runs. The degree of leaching is below the maximum level specified by the EN 12662 Standard Specification for Diesel Fuel Oils contamination content, i.e., 24 mg $L^{-1}$. This verifies that the catalyst used in this study has the potential to be resilient to leaching and demonstrates robust stability. $NH_3$-TPD was then utilised in order to determine the properties of the spent catalyst (Figure 7b). The data from this analysis revealed that there was a decrease in both catalyst acidity and acidity strength owing to the leaching of elements. Overall, the reduction of acidic sites and their strength demonstrated the importance of the $Mn_{(0.5\%)}$-$Mo_{(0.5\%)}$/AC catalyst's acidic properties in changing the relative rates of the deCOx mechanism, resulting in the formation of diesel range fuel. Evidently, a lack of acidic sites reduced the chances of C–O bond cleavage activity yet enhanced C–C bond cleavage, thereby yielding greater light fractions. This was in agreement with the reduction of the desired *n*-$C_{15}$+$C_{17}$ fractions on the 5th run. The decarboxylation/decarbonation ratio was evaluated using Thermal Conductivity Detector-gas chromatography (GC-TCD) of the gases bioproduct for the first run and the fifth run at the optimum conditions. The GC-TCD results show that in the first run, the decarboxylation/decarbonation ratio was 31.98% and the decarboxylation was 66.2% for the decarbonation and about 1.82% for the thermal cracking. Additionally, the catalyst after the 5th run showed stability on the decarboxylation/decarbonation ratio with 29.93% for the decarboxylation ($CO_2$), 68.36% for the decarbonation (CO) and 1.71% ($CH_4$) for the thermal carking (Table S1).

### 2.5. Proposed Reaction Pathways for Deoxygenation of SPO via $Mn_{(0.5\%)}$-$Mo_{(0.5\%)}$/AC

Figure S3 depict the chemical equation for the catalytic deoxygenation of SPO as a summary of the current investigation. SPO was mostly made up of $C_{18}$ and $C_{16}$ fatty acid derivatives, according to its fatty acid composition profile. The carboxyl and carbonyl groups from $C_{18}$ and $C_{16}$ fatty acid derivatives are theoretically eliminated by deoxygenation, resulting in hydrocarbon fractions mostly constituted of *n*-heptadecenes (*n*-$C_{17}$) and

*n*-pentadecenes (*n*-$C_{15}$), as well as by-products ($CO_2$, CO and $H_2O$). Instead of mixes of *n*-$C_{15}$ and *n*-$C_{17}$, a large relative quantity of *n*-$C_{15}$ fraction was produced in this investigation, indicating that mild cracking is likely to occur and lead to C–C cleavage of the *n*-$C_{17}$ fraction moving into the *n*-$C_{15}$ fraction (by removing ethane). Bimetallic Mn-Mo phases are thought to enhance selected deoxygenation-mild cracking routes. The cracking ability of $Mn_{(0.5\%)}$-$Mo_{(0.5\%)}$/AC is due to the Mn and Mo, as shown in the cracking activity of Mn/AC and Mo/AC catalysts. Alternatively, C–C cleavage on fatty acid derivatives may occur, resulting in the creation of $C_{16}$ fatty acids, which then undergo a selective deoxygenation process to provide *n*-$C_{15}$ hydrocarbon compounds. Because the cracking process is inescapable, hydrocarbons frequently undergo C–C cleavage, resulting in short-chain hydrocarbon *n*-($C_7$–$C_{14}$) [32,33].

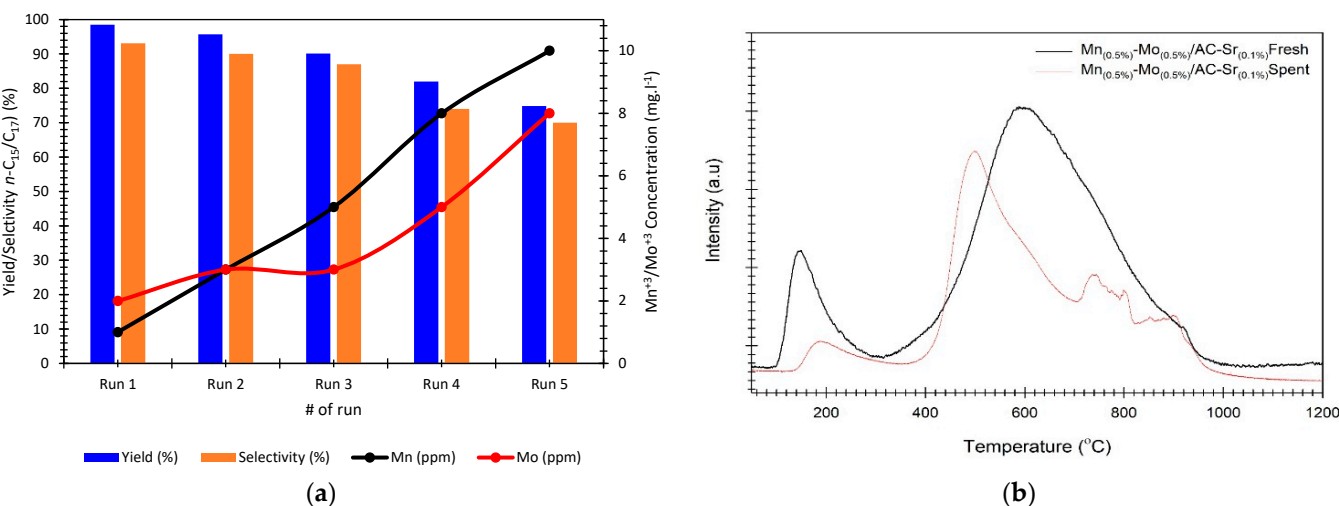

(**a**)                 (**b**)

**Figure 7.** Catalyst stability and reusability study; (**a**) the catalyst reusability yield/selectivity and active element leaching at optimum condition and (**b**) acidity measurements of fresh and spent supported catalyst of $Mn_{(0.5\%)}$-$Mo_{(0.5\%)}$/AC.

## 3. Materials and Method

### 3.1. Materials

Manganese (II) nitrate tetrahydrate ($Mn(NO_3)_2.4H_2O$), Molybdenum (II) acetate dimer ($Mo_2(OCOCH_3)_4$) and Strontium nitrate ($Sr(NO_3)_2$) with 99.0% purity was purchased from Merck (Germany); Citric acid ($HOC(COOH)(CH_2COOH)_2$) with 99.5% purity was purchased from Merck (Germany). The liquid products are alkane and alkene; hence, the standard of *n*-($C_7$–$C_{20}$) for gas chromatography analysis, and the internal standard used is 1-bromohexane. All the liquid standards were purchased from Sigma Aldrich and were utilised without further purification. For dilution, GC grade *n*-hexane with purity >98% (Merck, Darmstadt, Germany) was employed. $N_2$ gas (99% pure) was purchased from Linde Sdn. Bhd. Malaysia. Palm oil-based SPO (with FFA of 42.35%, Palmitic acid C16:0 of 45.68% Oleic acid, C18:1 of 40.19% were the major fatty acid composition) was used as the feedstock for the study, it was purchased from a restaurant at the Serdang, Selangor, Malaysia, and was employed for the reaction without any purification.

### 3.2. Catalyst Preparation

Palm pressed fibre (PPF) was rinsed in hot water in order to eradicate any dirt and additional contaminants and then dried in an oven. The desiccated PPF was then pulverised utilising the Pulverisette 4 vario-Planetary Mill. The main disc was set at 1200 rpm and the planet pair at 700 rpm; the duration of the milling procedure was 3 h. The powdered material was recovered and passed through a 200 μm laboratory test sieve (Endecotts, Ltd., London, UK) so as to procure the requisite grain. This was immersed in 10 mmol citric acid in ambient conditions for 6 h with ongoing agitation at 500 rpm. The resulting admixture was dried in a vacuum for 24 h at a temperature of 100 °C. The dried powder was introduced

into a horizontal tubular furnace employing a perforated cylindrical stainless-steel reactor with dimensions of $30 \times 10$ cm$^2$. The sample underwent pyrolysis at a temperature of 500 °C for 120 min using a water-steam flow rate with nitrogen of 100 cm$^3$ min$^{-1}$. Under continuing nitrogen steam flow, the temperature of the furnace was permitted to cool for a further 120 min. Then, the final product was washed with deionised water to pH 7 and dried in an electric oven, followed by the doping of an appropriate amount of Mn and Mo nanoparticles on the activated carbon. The process was decorated with Strontium(II) nitrate (as a promoter), Manganese (II) nitrate tetrahydrate and Molybdenum (II) acetate dimer according to the hot injection method [34,35]. The decoration was carried out in a Schlenk line setup in a 250 mL flask. In the process, 0.4 mL oleic acid, 0.4 mL oleylamine, 6.86 mg 1,2-hexadecanediol and 20 mL diphenyl ether were purged by vacuum and then charged with nitrogen and repeated 5 times. The mixture was then degassed together under vacuum for a period of 1 h at a temperature of 85 °C. The mixture was later placed under an argon atmosphere and heated to a temperature of 200 °C. Meanwhile, 5 g of AC were dispersed in dichlorobenzene and mixed with the required amount of Manganese (II) nitrate tetrahydrate and Molybdenum (II) acetate dimer and heated under sonication to 80 °C. The Mn-Mo/AC mixture was then injected into the three-neck flask and stirred for 10 min. The reaction was stopped, and the mixture was cooled in a water bath. The mixture was subsequently washed in a three-fold excess of MeOH and centrifuged (3500 rpm, 10 min) to remove the non-reacted chemicals and non-attached Mn and Mo nanoparticles. The supernatant was removed, and the resulting Mn-Mo/AC heterostructure was re-dispersed in toluene solution, which allowed the ligand exchange to occur.

Both the decorated Mn-Mo and the activated carbon were then transferred to an aqueous environment following a ligand exchange procedure. Typically, 5 g of the activated carbon were dispersed in 50 mL of toluene. In the meantime, 40 mL of anhydrous MeOH were mixed with 0.5 g of 2-mercaptopropionic acid (MPA). NaOH pellets were also added under the stirring condition until the pH was raised to above 10.5. The dispersion was then precipitated with acetone/MeOH (1:1, threefold excess) twice, and the new hydrophilic nanocrystals were mixed with toluene (10 mL) and precipitated to wash away the previous ligands (successfully ligand-exchanged nanocrystals are not dispersed). After disposing of the 4 supernatants and drying under N$_2$, the crystals were readily dispersed in deionised water. Ultimately, the samples were filtered through a 0.45 μm pore filter and the residual reactants were evaporated (rotation evaporator ~40 °C) and purged with N$_2$.

*3.3. Catalyst Characterisation*

An X-ray diffraction (XRD) technique (Shimadzu, model XRD-6000, Kyoto, Japan) was employed in order to determine the chemical configuration and the dispersion status of the metal-doped AC catalysts prior to and following the chemical reaction [36]. The surface area measurement of the pulverised specimen was assessed utilising a nitrogen (N$_2$) adsorption/desorption analyser (Thermo-Finnigan Sorpmatic 1990 series, San Jose, CA, USA) via the Brunauer–Emmett–Teller (BET) method. Pore size and volume distribution of the specimens of produced catalyst were additionally appraised. In general, the catalyst specimen was degassed overnight at a temperature of 150 °C in order to eradicate any alien volatile materials and water from its veneer. The true analysis was conducted at a temperature of −196 °C. A vacuum chamber was used for the N$_2$ desorption and adsorption methods on the catalyst surfaces [31].

The acidic and basic characteristics of the catalyst were assayed using the temperature-programmed desorption (TPD) process. A pair of probe molecules, i.e., NH$_3$ for acid (TPD–NH$_3$), was utilised. The procedure was performed using a Thermo-Finnigan TPDRO 1100 model, which included a thermal conductivity detector (TCD). Approximately 50 mg of catalyst were pre-treated with N$_2$ gas flow for 30 min at a temperature of 250 °C. Subsequently, the catalyst was exposed to NH$_3$ gas adsorption for 60 min at a temperature of 50 °C, and then the TCD was employed in order to recognise the NH$_3$ desorption from the catalyst basic locus with a stream of helium gas at a rate of 30 mL/min between

temperatures of 50 °C and 900 °C. The temperature was maintained at a steady level for 30 min. [37]. A LEO 1455 VP was used for the recording of the field emission scanning electron microscopy (FESEM) data.

A Perkin–Elmer Emission spectrometer model Plasma 100 was utilised for the inductively coupled plasma-atomic emission spectrometry (ICP-AES) analysis in order to establish the element constitutions of the specimens of catalyst, i.e., Mn and Mo [38].

### 3.4. Catalytic Activity Evaluation

SPO′s DO reactions were performed in a 100 mL magnetically agitated batch reactor. Approximately 10 g of SPO were inserted into the reactor together with 0.1 wt.% of catalyst for each experiment. Before commencing the study, the reactor was cleared with inert gaseous $N_2$, and the admixture was repeatedly stirred so as to subtract surplus oxygen from the specimen. The combined compounds underwent unremitting agitation in the ongoing inert $N_2$ flow, the latter at 20 cc/min at atmospheric pressure in order to make certain that the extraction of oxygen was maintained throughout the DO reaction.

The temperature was gradually increased until the goal temperature of 300 °C was reached; the latter was sustained for 60 min. The condensation of relevant substances was achieved by the utilisation of a cooler. These were stored within a container within the batch reactor, which itself was permitted to cool to ambient temperature at the conclusion of the individual experiments via a cooling system reliant on the exchange of extrinsic water. The ultimate liquid products were analysed using gas chromatography, a flame ionisation detector (GC-FID), Fourier-transform infrared spectroscopy and gas chromatography-mass spectrometry (GC-MS).

### 3.5. Product Analysis

The resultant liquid products underwent quantitative appraisal utilising a Shimadzu GC-14B gas chromatograph, which was made up of an HP-5 capillary column 30 m in length, inner diameter and film thickness of 0.32 mm and 25 μm, respectively, and an FID, which operated at 300 °C.

The liquid products were diluted before conversion analysis with GC grade *n*-hexane and 1 μL of the specimen before conversion analysis assessment. The solution was then placed into the GC column. The temperature of the injection was maintained at 250 °C; the carrier gas selected was $N_2$. The oven′s true temperature was kept at 40 °C for 6 min; it was then turned up to 270 °C at a rate of 7 °C per min.

1-bromohexane was employed as an intrinsic reference for the quantitative assessments. Work conducted prior to the current experiments using CG-MS for the analysis of the liquid products had identified saturated and unsaturated hydrocarbon fractions, i.e., $C_7$–$C_{20}$. The latter was, therefore, additionally characterised by GC-FID analysis utilising reference substances from industrial saturated hydrocarbons within the $C_7$–$C_{20}$ spectrum together with purchased unsaturated nonene [39].

### 4. Conclusions

In this study, the influence of diverse catalyst configurations on both deactivation and harvest were examined in depth through the DO of a number of catalysts, such as $Mn_{(0.5\%)}$-$Mo_{(0.5\%)}$/AC. SPO was the reactant feedstock utilised, which had a significant acid value; FFA components were over 95%. Both saturated and unsaturated fatty acid units made up the feedstock oil. The data suggest that the feedstock underwent efficacious DO with all three forms of catalyst; oxygen was extracted to yield carbon monoxide, carbon dioxide and water. The bimetallic catalyst shows a great effect on the catalyst activity due to the synergic effect between both the active element (Mn and Mo). An optimum DO condition for SPO over $Mn_{(0.5\%)}$-$Mo_{(0.5\%)}$/AC catalysts is by using a 2 wt.% catalyst and a temperature of 330 °C for over an hour with hydrocarbon yield of 96.37% and 94.44% *n*-$C_{15}$+$C_{17}$ selectivity. The $Mn_{(0.5\%)}$-$Mo_{(0.5\%)}$/AC catalyst showed excellent catalyst stability since it can be used for up to five serial runs with a hydrocarbon yield of >72%

and $n$-$C_{15}$+$C_{17}$ products with a selectivity of >70%. It should be noted that the catalyst activity effectiveness reduces by reaction runs, owing to the loss of the active metals into the solution during serial runs.

**Supplementary Materials:** The following supporting information can be downloaded at: https://www.mdpi.com/article/10.3390/catal12050566/s1, Table S1: The biproduct gas analysis using Thermal Conductivity Detector-gas chromatography (GC-TCD) at the optimum condition using Mn (0.5%)-Mo (0.5%)/AC; Figure S1: 2-Catalyst evaluation using different feedstocks (Sludge palm oil and fresh palm oil); Figure S2: 3-Deoxygenation reaction mechanism of unsaturated and saturated free fatty acids.

**Author Contributions:** Conceptualization, A.G.A.; Data curation, A.I. and N.M.; Formal analysis, A.I., S.H.T. and M.F.N.; Funding acquisition, A.G.A.; Investigation, S.Z.R. and Y.H.T.-Y.; Methodology, S.M.; Project administration, S.Z.R.; Software, A.G.A.; Supervision, A.G.A. and Y.H.T.-Y.; Validation, L.K.O.; Writing—original draft, A.G.A.; Writing—review & editing, N.A.-M. All authors have read and agreed to the published version of the manuscript.

**Funding:** The authors acknowledge the financial support from Ministry of Higher Education Malaysia for Fundamental Research Grant Scheme (FRGS-MRSA/1/2019/STG01/UPM/01/30), Galakan Penyelidik Muda (GGPM) (GGPM-2020-015) and GP-2021-K023310.

**Data Availability Statement:** The data that support the findings of this study are available from the corresponding author, [Abdulkareem Ghassan Alsultan], upon reasonable request.

**Conflicts of Interest:** The authors declare no conflict of interest.

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
