# Peer review of "Selective Deoxygenation of Sludge Palm Oil into Diesel Range Fuel over Mn-Mo Supported on Activated Carbon Catalyst"

_catalysts, doi:10.3390/catal12050566_

Round 1
Reviewer 1 Report
The paper by Yap et al., deals with the synthesis of green diesel by the use of a bimetallic catalyst on activated carbon. To produce fuel, the authors employ sludge palm oil as a raw material for the process.
The recycling of the catalyst has also been studied and 5 tests can be carried out.The paper can be accepted for the publication after some minor revisions, listed below:
1) The paper should be revised asthere are some errors;
2) In the abstract, silver is cited, but it seems a mistake;
3) The intro section is too long and it should be revised;
4) Pg3, results and discussion: "The dimensions of the particles were diminished... from 247 to 293 nm" (!). Please, check.
5) Pg. 4, results and discussion: "(2a), suggesting that cerium...into the pore (Figure 2)". Why cerium is there? Where is it from?
In conclusion, even if the preparation of the catalyst is too long and tedious and its use is limiteted to 5 runs, results are sufficient and the research is well conducted. As a result, the paper can be published on Catalyst, after the revision and the acceptance of the suggestions reported above.
Author Response
Thank you for your kind comments and your time to improve our article. All comment were considered.

Reviewer 2 Report
This paper investigated the selective deoxygenation of sludge palm oil into diesel range with different Mn-Mo/AC catalyst with different Cu to Mo ratio. It was an interesting work and provided valuable data for bio-refinery. but, there are some questions as below:
- In Fig.4, why the Mo/AC catalyst was not evaluated? Please add this section.
- What about the decarbonylation and decarboxylation ratio, is not reflected in the article.
- The Mn(0.5%)-Mo(0.5%)/AC catalyst has the best selectivity and hydrocarbon yield, what kind of synergistic mechanism leads to this? What is the connection between the reaction process of oil to hydrocarbons and the nature of the catalyst? Is the valence bond of Mo changing (Mo4+ to Mo6+) or something else? Please add the XPS to confirm this part.
- It is puzzling how Sr and Ce came from in AC support “The X-ray diffraction (XRD) pattern of AC-Sr evidenced no crystalline Sr2O phases (2a), suggesting that cerium is either highly dispersed on the AC external surface or doped into the pore (Figure 2).”
- Author should consider the calculation of deoxygenation degree of palm oil to evaluate the catalytic pyrolysis performance, if possible.
- Would the authors be able to provide some extended discussion on the fundamental reaction mechanisms using Saturated and unsaturated fatty acids? What are the detailed dominant reactions and what are their contributions to the overall reactions?
Author Response

(The authors gave the same response as above.)

Round 2
Reviewer 2 Report
Accept